# A method for detecting the quality of cotton seeds based on an improved ResNet50 model

Xinwu Du[1,2]*, Laiqiang Si[1], Pengfei Li[1], Zhihao Yun[1]

1 College of Agricultural Equipment Engineering, Henan University of Science and Technology, Luoyang, Henan, China, 2 Science & Technology Innovation Center for Completed Set Equipment, Longmen Laboratory, Luoyang, Henan, China

☉ These authors contributed equally to this work.

* du_xinwu@sina.com.cn

**Data Availability Statement:** All relevant data are within the manuscript and its Supporting Information files.

**Funding:** Funding:National Natural Science Foundation of China (52075150), Natural Science

## Abstract

The accurate and rapid detection of cotton seed quality is crucial for safeguarding cotton cultivation. To increase the accuracy and efficiency of cotton seed detection, a deep learning model, which was called the improved ResNet50 (Impro-ResNet50), was used to detect cotton seed quality. First, the convolutional block attention module (CBAM) was embedded into the ResNet50 model to allow the model to learn both the vital channel information and spatial location information of the image, thereby enhancing the model's feature extraction capability and robustness. The model's fully connected layer was then modified to accommodate the cotton seed quality detection task. An improved LRelu-Softplus activation function was implemented to facilitate the rapid and straightforward quantification of the model training procedure. Transfer learning and the Adam optimization algorithm were used to train the model to reduce the number of parameters and accelerate the model's convergence. Finally, 4419 images of cotton seeds were collected for training models under controlled conditions. Experimental results demonstrated that the Impro-ResNet50 model could achieve an average detection accuracy of 97.23% and process a single image in 0.11s. Compared with Squeeze-and-Excitation Networks (SE) and Coordination Attention (CA), the model's feature extraction capability was superior. At the same time, compared with classical models such as AlexNet, VGG16, GoogLeNet, EfficientNet, and ResNet18, this model had superior detection accuracy and complexity balances. The results indicate that the Impro-ResNet50 model has a high detection accuracy and a short recognition time, which meet the requirements for accurate and rapid detection of cotton seed quality.

## 1. Introduction

Cotton seed is the foundation of cotton production, and its quality directly impacts cotton yield and quality [1, 2]. The quality of cotton seed is under increasing scrutiny as the mechanized one-hole, one-seed precision sowing technology becomes more prevalent in China [3–5]. Phenotypic defects are one of the criteria for evaluating the quality of cotton seed. Cotton seed defects are traditionally detected manually, which is laborious, time-consuming, and subjective. Therefore, developing an objective and automated method for detecting cotton seeds is necessary.

Foundation of Henan Province (No.202300410124) and Guangdong Key R&D Program (No.2019B020222004).

Machine learning-based image processing techniques have been successfully applied to detect seed quality with the advancement of computer vision technology [6–8]. The researchers conduct seed quality assessment by extracting features such as texture, color and shape of the seed images. This method is more advanced and effective in detecting seed quality than the manual method. However, the method is relatively dependent on manual feature extraction, and different features require different extraction methods. In addition, manual feature extraction is usually inadequate. Thus, it leads to the detection accuracy of the method is not high.

There has been an increase in convolutional neural networks (CNN) used for image recognition [9–11]. In addition to simulating the human brain's mechanism for extracting features in layers, the technique can extract features automatically from simple to complex, from bottom to top, and from concrete to abstract. Several researchers have successfully applied CNN to the detection of seed quality [12–15]. But, a disadvantage of CNN detection is that it requires a large amount of training data, is time-consuming, and is computationally resource-intensive.

To address the shortcomings of existing methods, this paper proposes a new CNN for cotton seed quality detection. A summary of this study's major contributions and innovations is provided below.

1. Based on the appearance of defects in cotton seed, a new cotton seed dataset is created to support the development of subsequent detection algorithms.

2. The Impro-ResNet50 model is proposed as a new method for detecting cotton seed quality based on an attention mechanism. The CBAM attention block is embedded in ResNet50 to integrate feature channel and spatial information attention and enhance the model's capacity to learn essential information about cotton seed regions.

3. The model's application serves as a reference for developing new models, demonstrating the interoperability of deep learning models and attention mechanisms.

4. On the basis of the cotton seed quality identification dataset, Impro-ResNet50 is subjected to extensive comparative experiments. Impro-ResNet50 is highly accurate and robust in cotton seed detection tasks, demonstrating the efficacy of the CBAM module. Provide technical support for developing cotton seed quality testing equipment in the future.

## 2. Related works

### 2.1. Application of machine vision technology to the detection of seed quality

The machine vision-based detection technology of seed quality has become relatively mature. Using image processing technology, the authors of [16] created an online detection system for soybean seeds. The system was based on classifying surface information such as the color, texture, and shape of soybeans and achieved a detection accuracy of over 97% for cracked and healthy soybeans. The authors of [17] chose high-quality pepper seeds using machine vision and classifiers. Multiple physical characteristics, such as the seeds' width, length, and projected area, were used as classification criteria. It detected high-quality and low-quality seeds with a greater than 90% accuracy. The authors of [18] described a low-rank Joint Multi-Modal bag-of-feature (JMBoF) classification method for detecting the appearance quality of soybean seeds. The model achieved 82.1% accuracy in detecting healthy, good, and unhealthy soybean seeds based on the color of the seeds. The authors of [19] combined spectral imaging and machine vision techniques to detect damage to sugar beet seeds. This method achieved a detection accuracy of 82% for five distinct types of sugar beet damage. In [20], the authors proposed

a machine vision-based, one-class classification method for evaluating the quality of tomato seeds. A 97% accuracy rate was achieved in classifying healthy and infected seeds. Combining automatic X-ray analysis and machine learning models, the authors of [21] presented a method for classifying the quality of Jatropha curcas seeds. The technique detected normal and abnormal seeds with a 94.36% accuracy rate. In [22], the authors developed a machine vision-based algorithm to detect moldy and normal maize seeds based on the difference in surface color, which had an overall detection accuracy of no less than 94%. In [23], the authors developed a machine vision-based double-sided rice seed identification system. The method identified rice seeds with open glumes using Hough linear detection and feature extraction. The algorithm achieved recognition accuracies of 88.1% and 87.7%, respectively, for normal and open rice seeds.

Although the above methods achieve excellent seed quality detection performance, it requires cumbersome image pre-processing and feature extraction. In addition, the input feature data limited the model's accuracy, which was often inadequate, resulting in poor detection accuracy.

## 2.2. Application of convolutional neural networks to the detection of seed quality

The CNN has started to be used to perform the quality detection work of seeds. For instance, the authors of [24] demonstrated a CNN-based transfer learning method for detecting haploid and diploid maize seeds. The model achieved optimal detection accuracy of 94.22%, providing technical support for the non-destructive, rapid, and inexpensive detection of high-quality seeds. In [25], the authors developed a peanut seed quality detection method based on machine vision and an adaptive CNN. The process achieved an average detection accuracy of 99.70% for common peanut seeds, such as mouldy, broken, or shrivelled. The authors of [26] integrated near-infrared hyperspectral imaging (NIR-HSI) and CNN deep learning techniques to differentiate between viable and inviable seeds. The process achieved a 90% detection rate for seeds. In [27], the authors presented an enhanced MobileNetV2-based model for detecting soybean seeds of superior quality. The detection accuracy of this model was 97.84%, which achieved the best results compared to the other seven models mentioned in the paper for detecting the quality of soybean. The authors of [28] claim that a photonic sensor based on laser backscattering and deep transfer learning was used to detect seeds of superior quality. The method achieved a 98.31% detection rate for high and low-quality seeds. Based on deep convolutional generative adversarial networks (DCGAN) and NIR-HSI, the authors of [29] proposed a method for identifying substandard wheat. In comparing support vector machine (SVM) and decision tree (DT) classifiers, the method demonstrated the best performance, with 96.67% detection accuracy for unsound wheat. Another study [30] presented a model for detecting maize seed defects based on a watershed algorithm and a dual-pathway CNN model. This method outperformed the conventional image processing techniques mentioned in the paper, with an average detection accuracy of 95.63% for both defective and healthy maize seeds.

Although seed quality detection has been extensively studied in previous research, there are currently no mature CNN-based detection models for cotton seed quality detection. Consequently, we anticipate that the proposal will address the current limitations of cotton seed quality detection models and reduce costs without compromising detection performance.

## 3. Experimental data

### 3.1. Data acquisition

GK-10 lazy cotton seeds harvested in 2021 were utilized as experimental material. This cotton seed variety was widely cultivated, high-yielding, disease-resistant, well-adapted, and

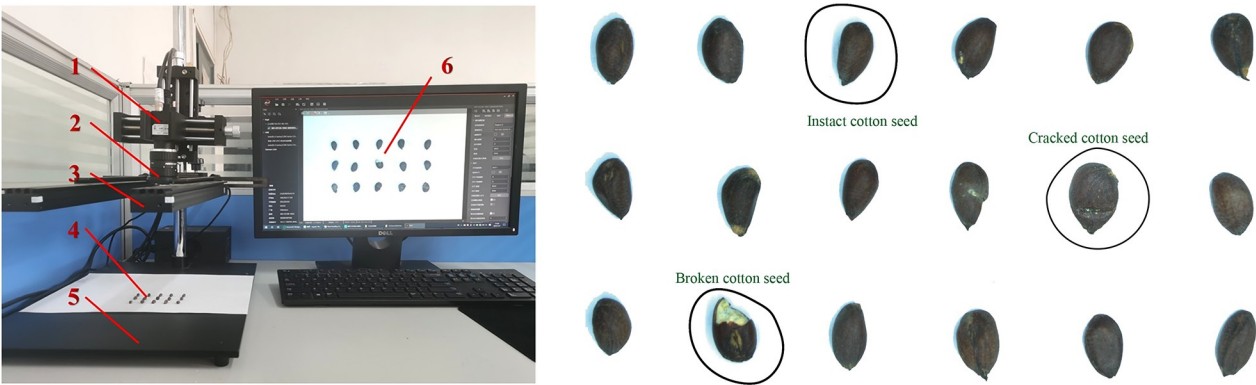

**Fig 1. Cotton seed acquisition.** (a) Image acquisition system: 1. Camera, 2. Lens, 3. Light-emitting diode (LED) lamps, 4. Cotton seed, 5. Platform, 6. Image monitor. (b) Cotton seed image.

representative. A random sample of 50 copies from the purchased material was taken, and 100 cotton seeds were randomly selected from each copy for integrity detection. The results showed that the proportion of cracked and partially broken cotton seeds in the material ranged from 5% to 10%. The phenotypic characteristics of the three types of cotton seeds were determined by observing intact, broken, and cracked cotton seeds in the material. Overall, intact cotton seeds were brown with entire edges and no discernible surface defects. Cracked cotton seeds were distinguished by surface cracks and a shift in color gradation at the cracks. Broken cotton seeds exposed the milky white endosperm at their edges.

In the indoor environment (Natural Light + energy-efficient lamp), each batch of 18 seeds was distributed randomly in a 36 pattern. The seed samples were photographed vertically from 20 to 25 cm using a Hikvision CCD (MV-CE200-10UC model) camera and 12 mm lens (MVL-HF1224M-10MP model) with an image resolution of 4024×3072 pixels. 3154 images were acquired in total. The image acquisition system is shown in Fig 1.

The image of single cotton seed was produced by cropping the entire picture. To meet the image input requirements of the CNN, the cotton seed image was uniformly scaled to 224×224 pixels. The individual seed images and the corresponding decomposition background images are shown in Fig 2. A total of 4419 images of cotton seeds were obtained, consisting of 1367 intact seeds, 1467 cracked seeds, and 1585 broken seeds.

## 3.2. Data preprocessing

To improve the model's generalisation and robustness, the data were expanded by flipping, rotating, scaling, cropping, panning, and adding noise to the three image types of cotton seed. The expanded cotton seed image dataset consisted of 7386 images, and the dataset was divided into 80% training set, and 20% validation set randomly using the Python program. The sample distribution of cotton seeds is shown in Table 1 (S1 Data).

## 4. Methodology

In cotton cultivation, low-quality cotton seeds lead to a reduction in yield and quality. At this time, deep learning techniques can detect cotton seed quality early and avoid sowing low-quality cotton seeds. To effectively detect the quality of cotton seeds, a deep learning network based on residual structure and embedded attention mechanism was proposed in this paper.

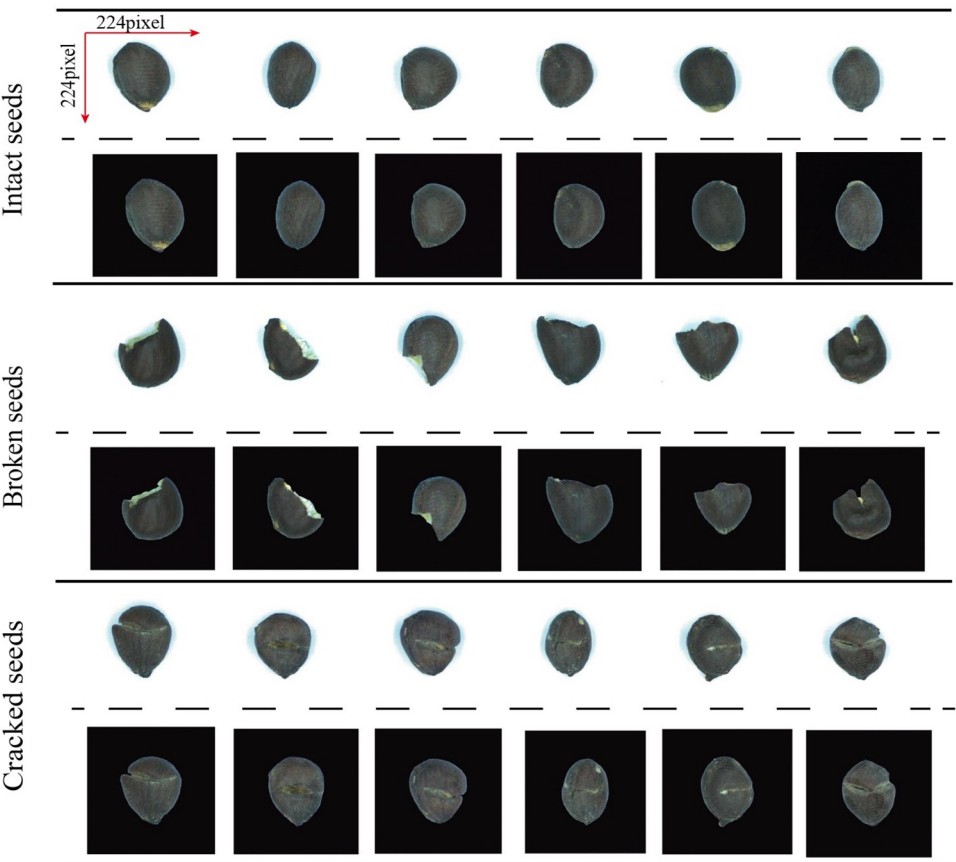

**Fig 2. Single cotton seed.**

## 4.1. Model improvement and structural design

**4.1.1. ResNet50 network structure.** When the CNN depth is increased, gradient degradation and disappearance will occur during the training process, resulting in difficulty in convergence and low accuracy [31, 32]. However, adding a residual structure to the CNN can largely avoid this phenomenon. The ResNet50 model and residual structure are shown in Fig 3.

The structure of the ResNet50 model is shown in Fig 3A. In Stage 1, the input image was reduced in size using a 7×7 convolutional layer and 3×3 maximum pooling downsampling. Then the higher-level features were extracted using the Conv2, Conv3, Conv4, and Conv5 residual structures in Stage 2. As a final step, the extracted high-dimensional features were fed into the fully-connected layer of Stage 3 for classification.

Two types of structures are available for the residual block, as shown in Fig 3B and 3c. Using 1×1 convolutional kernels before and after the 3×3 convolutional kernels to downscale and upscale could reduce the number of parameters in the model. Residual structure 3b was

**Table 1. Cotton seed data.**

| Category | Data set | Training set | Validation set |
|---|---|---|---|
| Intact seed | 2367 | 1894 | 473 |
| Broken seed | 2465 | 1972 | 493 |
| Cracked seed | 2554 | 2044 | 510 |

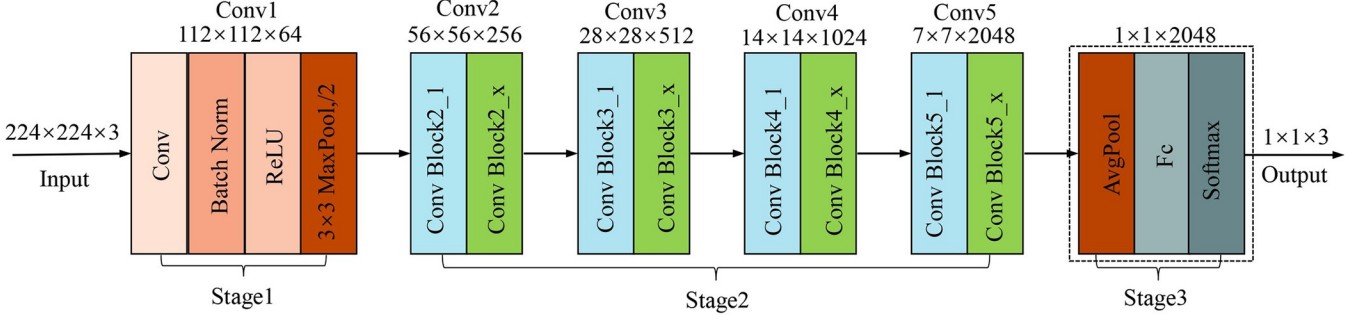

a. Network structure of classical Resnet50 model

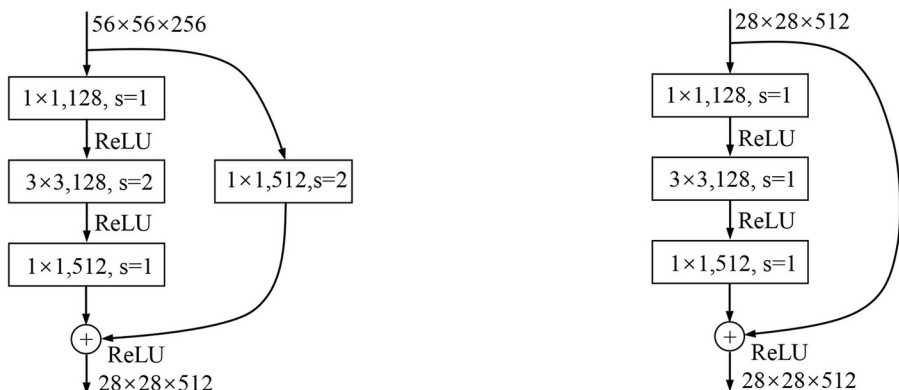

b. Residual structure with added scale

c. Residual structures that do not change dimensions

**Fig 3. The network structure of the Resnet50 model.**

the block with added scale, where the output feature matrix's height and width were half of the input through shortcut branching. This operation contributed to preventing model degradation. The residual structure 3c indicated that the feature size was unaltered, indicating that the output feature matrix's height and width were also unaltered.

**4.1.2. Convolutional block attention module model.** The CBAM module is a highly efficient attention module that can be incorporated quickly and flexibly into conventional classification networks without adding a large number of parameters, thereby enhancing the representation of features in convolutional neural networks [33, 34]. Using the CBAM module, The ResNet50 model could extract the features of cotton seed image channels while retaining the property of accurate spatial location information. The structure of CBAM is shown in Fig 4.

The Channel Attention Mechanism and Spatial Attention Mechanism made up the CBAM module. Given an input feature $F$, a channel compression operation was used to generate the channel attention weight $M_C$. Then, $M_C$ was multiplied by $F$ to obtain $F'$. The spatial attention weight $M_S$ was then generated by a 2-dimensional spatial compression operation and multiplied by $F'$ to produce $F''$. The specific calculation process is given in Eq 1.

$$\begin{cases} F' = M_C(F) \otimes F \\ F'' = M_S(F') \otimes F' \end{cases} \tag{1}$$

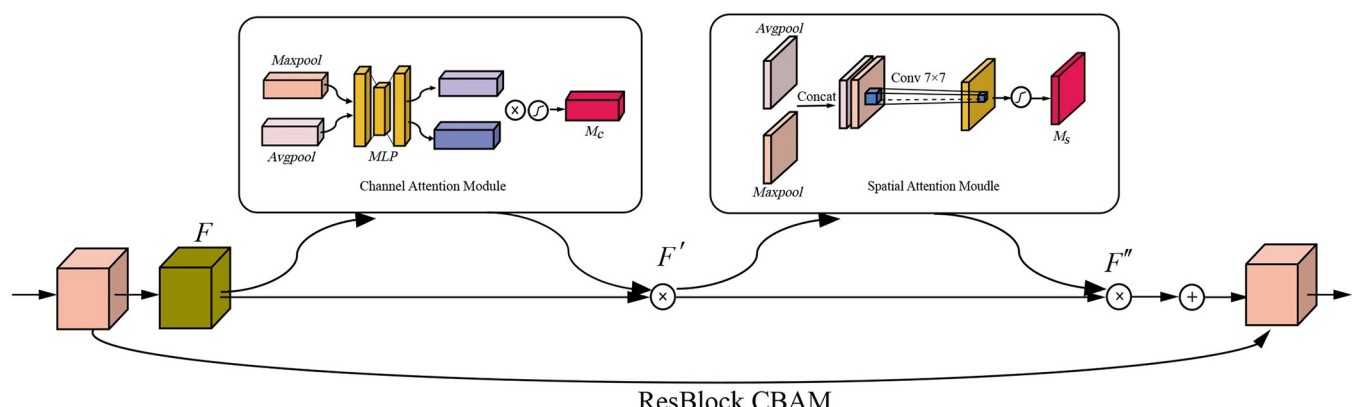

**Fig 4. CBAM module.**

where $F \in R^{C \times H \times W}$ represents the input feature matrix. $F' \in R^{C \times H \times W}$ represents the feature mapping selected by channel attention. $F''$ represents the feature mapping selected by spatial attention. $\otimes$ represents the element multiplication. $M_C \in R^{C \times 1 \times 1}$ and $M_S \in R^{1 \times H \times W}$ represents the channel attention weights, and the spatial attention weights, respectively. The calculations of $M_C$ and $M_S$ are given in Eqs 2 and 3.

$$M_C(F) = \sigma(MLP(AvgPool(F)) + MLP(MaxPool(F))) \tag{2}$$

$$M_S(F) = \sigma(f^{7 \times 7}([AvgPool(F); MaxPool(F)])) \tag{3}$$

where *MLP* is a two-layer fully connected neural network. $\sigma$ is the Sigmoid activation function. $f^{n \times n}$ is the convolution operation with a convolution kernel size of $n \times n$.

**4.1.3. Impro-ResNet50 model.** In this paper, the cotton seed detection model is based on the original ResNet50 network structure, but adds the CBAM attention mechanism after the Stage2 residual module and redesigned the fully connected layer and classification output layer. The Impro-ResNet50 model is shown in Fig 5.

The description of the cotton seed image detection procedure by Impro-ResNet50 is shown in Fig 5. The first step was converting a cotton seed input image to 224×224×3 pixels through pre-processing operations such as data enhancement and input into Impro-ResNet50. The residual block was then used to extract high-level characteristics from the image of cotton seed. By assigning greater weights to the most significant feature channels and smaller weights to the less substantial feature channels, the CBAM module was used to optimize the parts. As a consequence of the pre-convolution operation, the Impro-ResNet50 model would not lose any additional crucial information about cotton seeds due to the increased global attention. Finally,

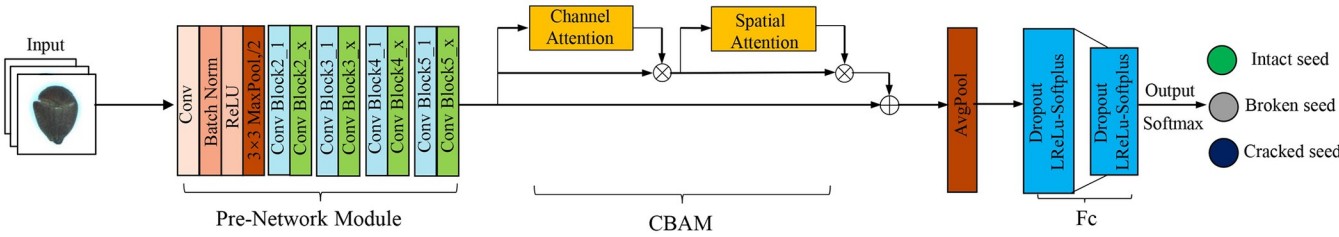

**Fig 5. The improved ResNet50 model.**

different classes of cotton seeds can be distinguished using the classifier pair.

## 4.2 Network training strategies

**4.2.1. Transfer learning.**   Transfer learning is applying knowledge learned in one source domain to another related target domain. Annotating large amounts of data in convolutional neural networks can be prevented, the model's dependence on data can be reduced, and the training efficiency of the model can be enhanced [35–37]. This study was motivated by this and trained the Impro-ResNet50 model using transfer learning.

ResNet50 was initially pre-trained on the massive public dataset ImageNet to obtain an initial converged weight in this study. This weight was then transferred to the Impro-ResNet50 model, which was trained using the previously self-constructed cotton seed dataset to generate new weights. Finally, the parameters of the Impro-ResNet50 model were fine-tuned to improve the model's learning performance for this dataset. Using transfer learning for weight initialization instead of random initialization of weights could accelerate the model's convergence and enhance its generalization capability.

**4.2.2. Activation function.**   The Relu activation function is widely utilized in CNN due to its quick operation and high performance. However, when the input was less than zero, the Relu activation could not continue to update the neuron death parameters. In the LRelu activation function, the activation value was determined by a threshold, and the parameters could continue to be updated if the input was less than 0. Although it addressed the issue of neural death, the LReLu function was not as smooth as the ReLu function. The Softplus activation function avoided the drawback of the Relu activation function's forced sparsity. Similar to the Relu function, it failed to address the function output offset phenomenon, negatively impacting the model's convergence performance [38–40]. The LRelu-Softplus activation function was designed by combining the characteristics of the three activation functions under appeal. The calculations of the four activation functions are given in Eqs 4 to 7.

$$f(x) = \begin{cases} 0, \ x \le 0 \\ x, \ x > 0 \end{cases} \tag{4}$$

where x ≤ 0, the output is 0, and the neuron is inactivated.

$$f(x) = \begin{cases} \alpha x, \ x \le 0 \\ x, \ x > 0 \end{cases} \tag{5}$$

where $\alpha = 0.01$, x<0, the output is negative, and the neuron is still active.

$$f(x) = \ln(e^x + 1) \tag{6}$$

$$f(x) = \begin{cases} \alpha x, & x \le 0 \\ \ln(e^x + 1) - \ln 2, & x > 0 \end{cases} \tag{7}$$

where $\alpha = 0.15$. The LReLu-Softplus activation function is shown in Fig 6.

**4.2.3. Optimisation algorithm.**   The optimizer is to update the network weights during the network training so that the model gets the optimal value. Adam is the most popular optimizer. The Adam algorithm estimates each gradient component's first and second-order moments to obtain the updated amount at each step and provides an adaptive learning rate

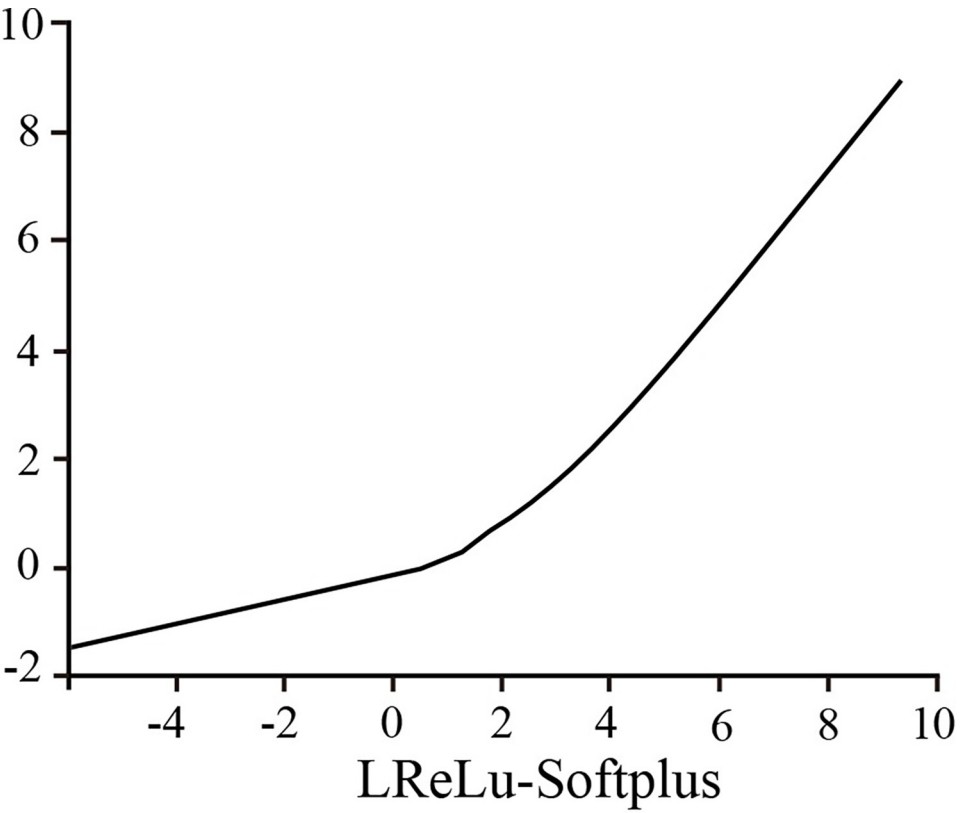

**Fig 6. LRelu-Softplus activation function.**

[41, 42]. The calculation of Adam's algorithm is given in Eq 8.

$$\begin{cases} g_t = \nabla_\theta f_t(\theta_{t-1}) \\ \hat{m}_t = \dfrac{\beta_1 \cdot m_{t-1} + (1 - \beta_1) \cdot g_t}{1 - \beta_1^t} \\ \hat{v}_t = \dfrac{\beta_2 \cdot v_{t-1} + (1 - \beta_2) \cdot g_t^2}{1 - \beta_2^t} \\ \theta_t = \theta_{t-1} - \alpha \cdot \dfrac{\hat{m}_t}{\sqrt{\hat{v}_t} + \varepsilon} \end{cases} \tag{8}$$

where $t$ is the number of time steps, $\theta_t$ is the update gradient. $g_t$ is the first-order derivative. $\beta_1$, $\beta_2 \in [0,1)$ is the exponential decay rate. $m_t$ is the estimate of the first-order moments, $\hat{m}_t$ is the bias-corrected estimate of the first-order moments. $v_t$ is the estimate of the second-order moments, $\hat{v}_t$ is the bias-corrected estimate of the second-order moments. $\alpha$ is the step size. $\varepsilon$ is an arbitrarily small positive number.

## 5. Experimental setup and evaluation indicators

### 5.1. Training platform and parameter settings

This test platform's software environment was a Windows 10 64-bit system with 16 GB of RAM. The CPU was an Intel Xeon E7, and the GPU was an NVIDIA GTX 1060. Pytorch used Python 3.8 as the programming language and Pytorch 1.9 as the deep learning framework to implement parallel processing of convolutional neural networks on the GPU.

**Table 2. Hyperparameters of Impro-ResNet50 model.**

| Parameters | Values |
|---|---|
| Optimizer | Adam |
| Learning rate | 1e−04 |
| Betas ($\beta_1$, $\beta_2$) | 0.9, 0.999 |
| Eps($\varepsilon$) | 1e−08 |
| Batch_size | 16 |
| Epochs | 300 |
| Dropout | 0.45 |
| Target_size | 224×224×3 |

The Adam optimization algorithm was chosen for the Impro-ResNet50 model with exponential decay rates of 0.9 and 0.999, respectively, and Eps of 1e−08. The convergence rate of the model was determined by the learning rate, which was set to 1e-04 in this study. Taking into account the training effect of the model and the experimental conditions, the batch size was set to 16, so 16 samples were entered into the model each time. To prevent overfitting, dropout was implemented before the final layer of the model to deactivate neurons with a predetermined probability, reduce the dependence between neurons, and enhance the model's ability to generalize. The dropout value was set to 0.45. The value of Epochs was set to 300. The images are then normalized before being fed into the CNN. The experiment's hyperparameters are shown in Table 2.

## 5.2. Network training process

Transfer learning was utilized in the training of the Impro-ResNet50 model. The training process of the model is shown in Fig 7. Initially, the cotton image dataset was loaded into the Pytorch deep learning framework and divided into training and validation sets using the dataset loading method. The ResNet50-pre.pth pre-training model should then be loaded. On the training set, the Impro-ResNet50 model was trained, and on the validation set, model evaluation results were obtained for each number of iterations. The cross-loss entropy function produced a gradual reduction in loss and increased precision. The model's training was concluded after 300 iterations, and the best training model was saved. Using the newly collected, unlabeled images of cotton seed, the best-trained model was identified, and the prediction results were presented.

## 5.3. Evaluation metrics

The confusion matrix's calculated accuracy ($A_{cc}$), precision ($P_r$), recall ($R_e$), and F1-score ($F_1$) were used as evaluation metrics in this study. Time spent processing a single image ($T_s$) was also crucial for evaluating models. Short training times for models are the solution to computational resource constraints [43–45]. The calculations of the five evaluation indicators are given in Eqs 9 to 13.

$$A_{cc} = \frac{T_p + T_n}{T_p + F_p + T_n + F_n} \tag{9}$$

$$Precision = \frac{T_p}{T_p + F_p} \tag{10}$$

$$Recall = \frac{T_p}{T_p + F_n} \tag{11}$$

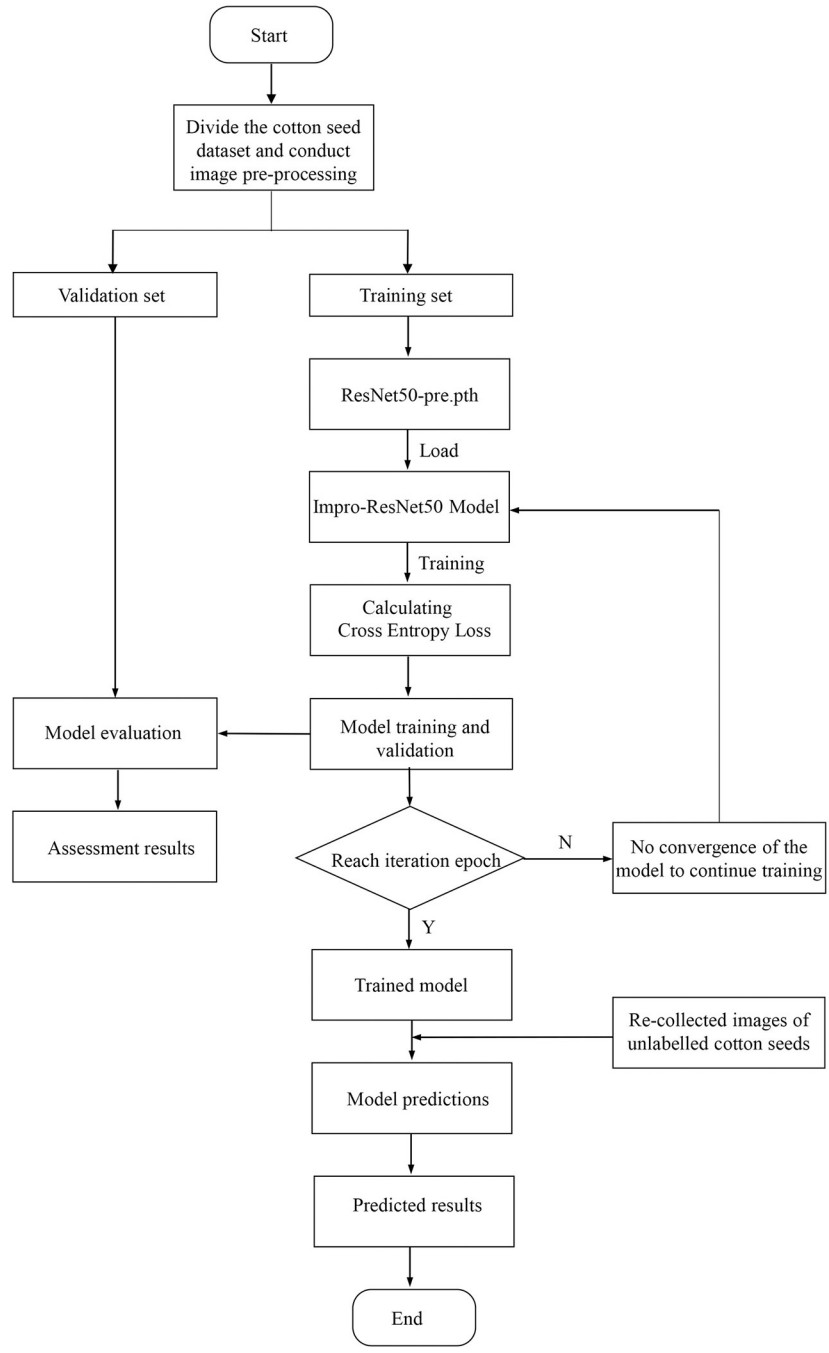

**Fig 7. Model training process.**

$$F1 = 2 \times \frac{Precision \times Recall}{Precision + Recall} \quad (12)$$

$$T_s = \frac{T}{N} \quad (13)$$

where $T_p$, $T_n$, $F_p$, and $F_n$ demonstrate the true positive, true negative, false positive, and false negative, respectively. $T$ is total train time. $N$ is the total number of train images.

## 6. Results

In this section, a re-collected dataset containing 450 unannotated images of intact, broken and cracked cotton seeds (150 of each type) was used for the performance evaluation of all models.

### 6.1. The impact of parameter optimization on the Impro-ResNet50 model

To determine the effects of the learning rate, activation function, and fully connected layer design on the performance of the Impro-ResNet50 model, the following three controlled experiments were designed. Experiment 1 compared the impact of various learning rates on the model's performance. Experiment 2 compared the impact of different activation functions on model performance. Experiment 3 compared the impact of fully connected layer layouts on model performance.

1. The impact of learning rate on the model.

The model converged slowly when Adam's optimization algorithm's learning rate was too low. A setting that was too large leads to non-convergence, and the loss function misses the optimal solution. The initial batch size was determined to be sixteen. In the Adam optimization algorithm, the default learning rate value was 0.001. To compare the training effect of the model, different orders of magnitude of parameter values were used, including 0.1, 0.01, 0.001, 0.0001 and 0.00001. The effect of learning rate adjustment on model loss values and accuracy is shown in Fig 8.

As shown in Fig 8, the model converges slowly, at a learning rate of 0.1. At a learning rate of 0.00001, the model barely converges, and the loss value is significant. When the learning rate was 0.01, 0.001, or 0.0001, the model converged well. However, when the learning rate was 0.0001, the model converged the quickest and had the smallest loss value after convergence. The model was tested with the highest accuracy at a learning rate of 0.0001 after 300 rounds of training. Consequently, the learning rate was set to 0.0001 during the Impro-ResNet50 model's training.

2. The impact of the activation function on the model.

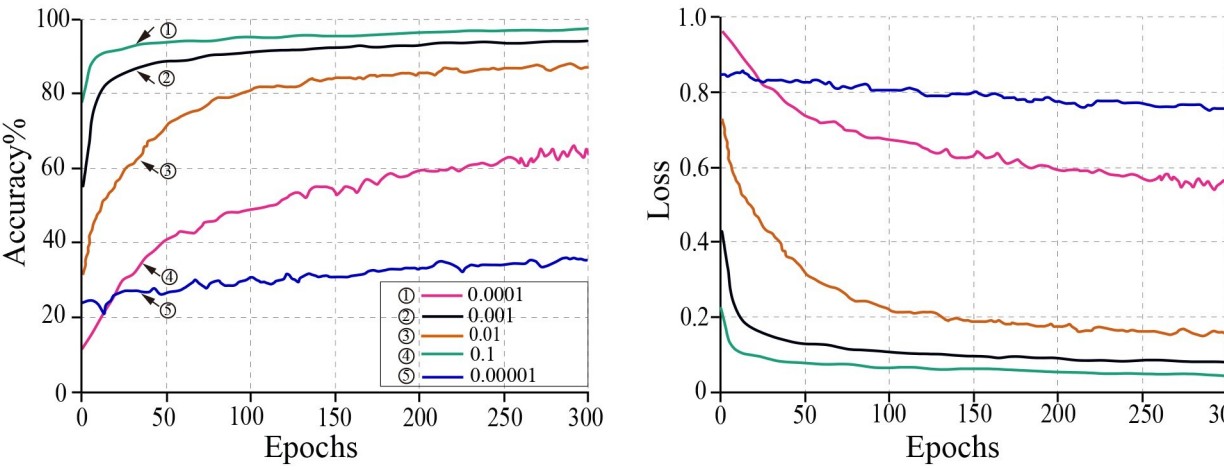

**Fig 8. Effects of learning rate on training accuracy and training loss.**

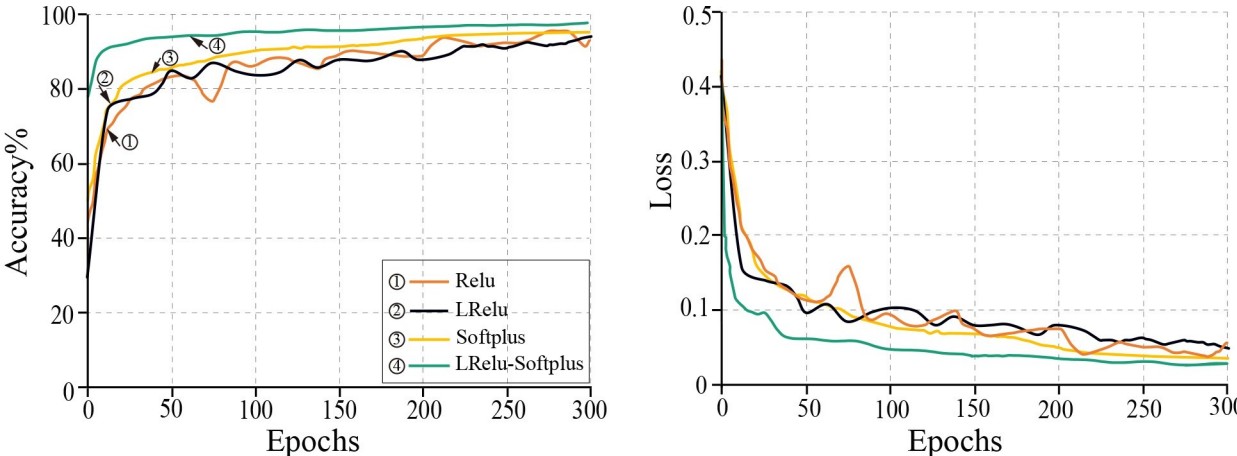

**Fig 9. Effects of activation function on training accuracy and training loss.**

The activation function played a crucial role in training the CNN, which provided the model with a robust capacity for fitting. The training effect of the model was compared when it was trained with Relu, LRelu, Softplus, and LRelu-Softplus activation functions using the fixed learning rate value of 0.0001. The effects of different activation functions on the loss values and accuracy of the model are shown in Fig 9.

As the number of iterations increased, the loss value of the model decreased, and the LRelu-Softplus activation function produced the fastest convergence and most stable training results, as shown in Fig 9. The other three activation functions exhibited more significant fluctuations in the training curve and larger loss values after the training was completed. Using the LRelu-Softplus activation function to train the model increased robustness and precision. The Impro-ResNet50 model was therefore trained using the LRelu-Softplus activation function.

3. Impact of fully connected layers on the model

In CNN, the fully connected layer combines the feature and classifier functions. Fully-connected layers contained a large number of model-size-affecting parameters. To determine the optimal number of fully connected model layers, the impact of adding one to three fully connected layers on model performance was compared. The effect of a different number of fully connected layers on the loss value and accuracy of the model is shown in Fig 10.

The model performs worst when configured with three fully connected layers, as shown in Fig 10. This could be due to the fact that the three-layer fully-connected layer resulted in excessive model parameters and overfitting during training. When comparing the model with one and two fully-connected layers, the model trained with two fully-connected layers had a smoother convergence and a lower loss value at the end of training. The Impro-ResNet50 model was therefore trained with two fully connected layers.

## 6.2. Effect of attention mechanism on model performance

To further validate the benefits of the model incorporating the CBAM attention mechanism, the CBAM attention mechanism was substituted with the ultra-lightweight attention mechanism models SE and CA for comparison experiments conducted under the same experimental conditions [46–49]. The experimental schemes were as described below. Scheme 1 model with no additional attention mechanism. Scheme 2 replaced the CBAM module for the SE module. Scheme 3 replaced the CBAM module for the CA module. Scheme 4 was the Impro-ResNet50 model of this paper. The results of the three experimental schemes are shown in Table 3.

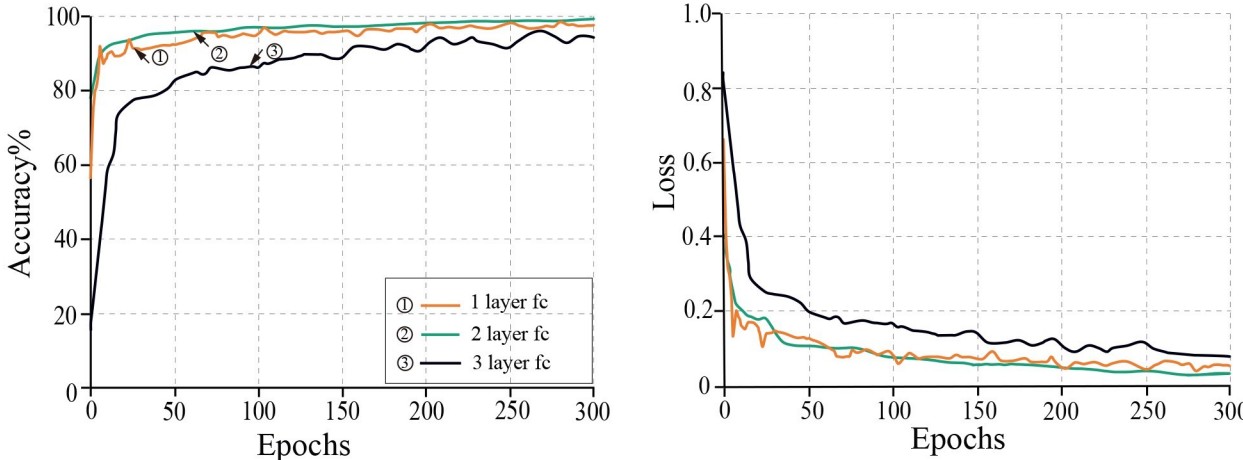

**Fig 10. Effects of fully connected layers on training accuracy and training loss.**

As shown in Table 3, the Impro-ResNet50 model achieved an average detection accuracy of 97.23% for cotton seeds, which was 0.82%, 1.11%, and 1.62% higher than the other three experimental models. The time required to detect a single image was 0.11s, which was 0.04s, 0.08s, and 0.38s faster than the other three experimental models. Comprehensive appeal results demonstrated the efficacy of introducing an attention mechanism to improve the model's accuracy. Moreover, only the SE model utilized the channel attention mechanism. In contrast, the other attention models presented in the paper were an organic combination of the channel attention mechanism and location feature data. Experiments comparing SE, CA, and CBAM attention revealed that spatial feature information contributed to the model's enhanced performance. In particular, the CBAM module improved model accuracy to the greatest extent, and the embedding of the CBAM module into the ResNet50 model enabled the model to simultaneously acquire channel information as well as spatial information about cotton seed regions, thereby improving the model's learning ability.

## 6.3. Performance comparison of other models

To further demonstrate the detection ability of the Impro-ResNet50 model, AlexNet, VGG16, GoogLeNet (InceptionV3), EfficientNet, and ResNet18 were chosen for migration learning and compared under identical experimental conditions. The experimental outcomes are shown in Table 4.

As shown in Table 4, the Impro-ResNet50 model outperformed the other five classical models in terms of detection accuracy and training time. The Impro-ResNet50 model detected cotton seeds with an average accuracy of 97.23%, which was 1.69%, 2.21%, 2.39%, 3.16%, and 5.05% higher than a variety of other models. In addition, the processing time for a single image was 0.11s, which was the fastest among all models. In the meantime, recall, precision,

**Table 3. Comparison of the results of cotton seed detection with different attention mechanisms.**

| No. | Model names | $P_r$/% | $R_e$/% | $F_1$/% | Params / M | $T_s$/s | $A_{cc}$/% |
|---|---|---|---|---|---|---|---|
| 1 | ResNet50 | 95.55 | 95.62 | 95.58 | 30.2 | 0.49 | 95.61 |
| 2 | Impro-ResNet50-SE | 96.00 | 96.05 | 96.02 | 31.5 | 0.19 | 96.12 |
| 3 | Impro-ResNet50-CA | 96.45 | 96.27 | 96.36 | 32.4 | 0.15 | 96.41 |
| 4 | Impro-ResNet50 | 97.33 | 97.13 | 97.22 | 33.4 | 0.11 | 97.23 |

**Table 4. Comparison of test results for different model cotton seeds.**

| Model names | $P_r$/% | $R_e$/% | $F_1$/% | Params / M | $T_s$/s | $A_{cc}$/% |
|---|---|---|---|---|---|---|
| AlexNet | 92.00 | 92.21 | 92.10 | 62.7 | 1.02 | 92.18 |
| VGG16 | 94.00 | 94.09 | 94.04 | 145.2 | 0.87 | 94.07 |
| GoogLeNet | 94.44 | 94.52 | 94.48 | 24.6 | 0.62 | 94.84 |
| EfficientNet | 95.11 | 94.99 | 95.02 | 42.6 | 0.21 | 95.02 |
| ResNet18 | 95.33 | 95.39 | 95.35 | 15.2 | 0.18 | 95.54 |
| Impro-ResNet50 | 97.33 | 97.13 | 97.22 | 33.4 | 0.11 | 97.23 |

and F1 all achieved positive outcomes. Although the number of parameters was not the lowest, it was within the affordability range for hardware. In terms of overall performance, the advantages of the model proposed in this paper were greater. The AlexNet model required the most time and had the lowest detection accuracy among the five classical models. The VGG16 model had the most parameters and required a substantial amount of computational resources. The GoogLeNet model implemented the Inception structure, which drastically reduced the number of model parameters and improved detection performance. EfficientNet utilized NAS technology to simultaneously search and optimize the model depth, width, and input image resolution in order to extend the model structure proportionally and attain a high level of structural proportionality. Therefore, the detection task also yielded good results. The ResNet18 model had a similar structure to the Impro-ResNet50 model, utilizing the residual structure to enhance its feature learning capability. However, its residual block consisted of 18 layers. Although the parameters were reduced compared to the Impro-ResNet50 model, the time consumption and detection accuracy in a single image were also diminished. The improved Impro-ResNet50 model could detect images of cotton seeds with greater precision.

## 6.4. Confusion matrix to visualize and analyse model detection results

A confusion matrix is a valuable tool for evaluating the quality of a classification model and its performance. Each row represents the actual data for a category, while each column represents the predicted data for that category, with the diagonal values indicating the likelihood of being accurate. The confusion matrix of the Impro-ResNet50 model, as shown in Fig 11.

As shown in Fig 11, the average classification accuracy of the model was 97.23%, and the classification performance (in terms of F1 score) for broken, intact, and cracked cotton seeds decreased from highest to lowest. There were 147 correct identifications out of 150 intact cotton seeds, 146 correct identifications out of 150 broken cotton seeds, and 145 correct identifications out of 150 cracked cotton seeds. By analyzing the misclassified images, it was determined that the Impro-ResNet50 model had a high misclassification rate when classifying cracked cotton seeds as intact cotton seeds. It was difficult for the model to detect cracked features in the images because they were dark, the overall resolution was low, and factors such as the angle of the shot made it difficult for the model to detect them.

## 7. Conclusion and future work

The construction of an attention-based mechanism for the cotton seed quality detection model. Integrating feature channels and spatial location information was accomplished by incorporating a CBAM module. A modified LRelu-Softplus activation function was used to enhance the model's capacity for generalization. The transfer learning strategy and Adam optimization training algorithm decreased model parameters and accelerated model convergence speed. The influence of parameter settings and attention mechanisms on the model was

**Confusion matrix**

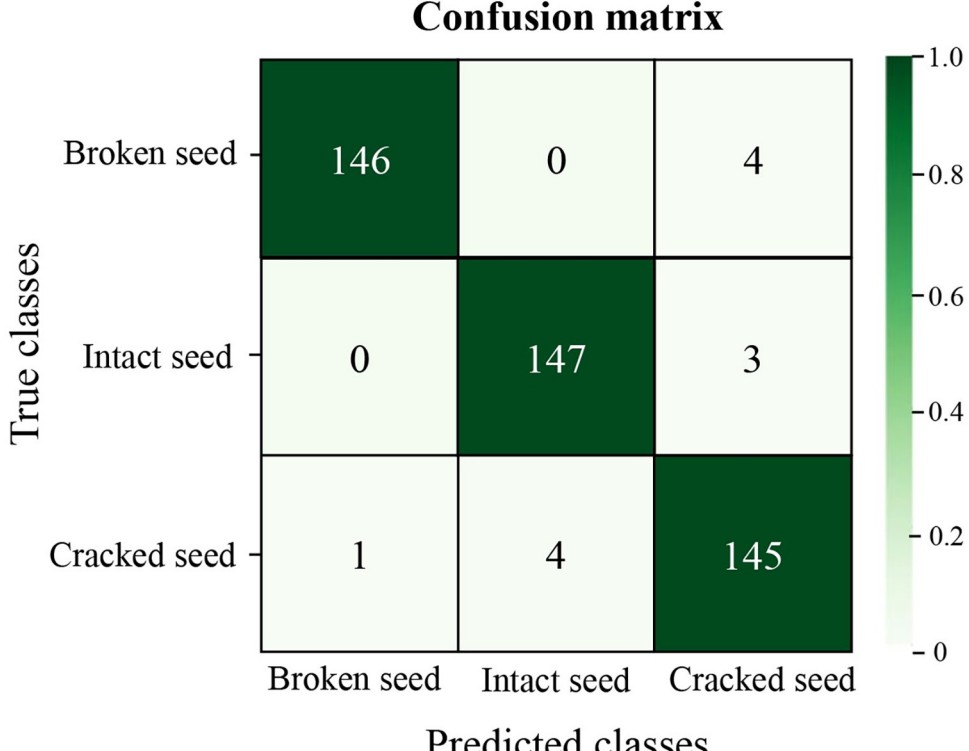

**Fig 11. Confusion matrix for the Impro-ResNet50 model.**

discussed and compared with AlexNet, VGG16, GoogLeNet, EfficientNet, and ResNet18. The following are the conclusions:

1. The Impro-ResNet50 model constructed with a learning rate of 0.0001, an activation function of LRelu-Softplus, and two fully connected layers converged the quickest and were the most robust. After training, the average detection accuracy of the Impro-ResNet50 model reached 97.23%, and the time required to process a single image was only 0.11s.

2. Compared to the three models without an embedded attention mechanism, the embedded SE attention mechanism, and the embedded CA attention mechanism, the average detection accuracy was improved by 0.82%, 1.11%, and 1.62%, respectively. The processing time of a single image was enhanced by 0.04s, 0.08s, and 0.38s, respectively, under identical experimental conditions.

3. Compared to traditional models such as AlexNet, VGG16, GoogleNet, EfficientNet, and ResNet18, the average detection accuracy was increased by 1.69–5.05% and the time required to process a single image was decreased by 0.07–0.91s.

4. The confusion matrix revealed that the Impro-ResNet50 model had a higher overall recognition accuracy and produced superior results for cotton seeds. However, the model still has a certain misclassification rate, with the detection of cracked cotton seeds performing the worst. Future research will be conducted on detecting cracked cotton seeds with less obvious taxonomic features.

The Impro-ResNet50 cotton seed quality detection model, based on the attention mechanism, was trained on a large amount of data while maintaining high accuracy and requiring

only a short amount of time to run. In the future, we will supplement the data with cotton seeds of different qualities and backgrounds so that the model has a wider range of applications. At the same time, the model is simplified so that it can be deployed on mobile and easily used by farmers.

## Supporting information

**S1 Data.**
(ZIP)

## Author Contributions

**Conceptualization:** Xinwu Du.

**Data curation:** Pengfei Li, Zhihao Yun.

**Funding acquisition:** Xinwu Du.

**Investigation:** Laiqiang Si.

**Methodology:** Xinwu Du, Laiqiang Si, Pengfei Li, Zhihao Yun.

**Project administration:** Xinwu Du, Laiqiang Si, Pengfei Li, Zhihao Yun.

**Resources:** Xinwu Du, Laiqiang Si.

**Software:** Xinwu Du, Laiqiang Si.

**Supervision:** Xinwu Du.

**Validation:** Xinwu Du.

**Visualization:** Xinwu Du.

**Writing – original draft:** Laiqiang Si.

**Writing – review & editing:** Xinwu Du.

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
