## [Decision Letter · Decision Letter 0]

14 Apr 2022

PONE-D-22-02418Cotton seed quality identification method based on improved ResNet50 modelPLOS ONE

Dear Dr. Du,

Thank you for submitting your manuscript to PLOS ONE. After careful consideration, we feel that it has merit but does not fully meet PLOS ONE’s publication criteria as it currently stands. Therefore, we invite you to submit a revised version of the manuscript that addresses the points raised during the review process.

We look forward to receiving your revised manuscript.

Kind regards,

Sathishkumar V E

Academic Editor

PLOS ONE

Journal Requirements:

Additional Editor Comments:

This kind of manuscript gives me a very good reading and review manage experience. More data analysis is needed to explain the screening of cotton seeds. Some details are not well explained. It will cause some confuse for readers.

Reviewers' comments:

Reviewer's Responses to Questions

**Comments to the Author**

1. Is the manuscript technically sound, and do the data support the conclusions?

Reviewer #1: Partly

Reviewer #2: Yes

2. Has the statistical analysis been performed appropriately and rigorously? 

Reviewer #1: No

Reviewer #2: I Don't Know

3. Have the authors made all data underlying the findings in their manuscript fully available?

Reviewer #1: Yes

Reviewer #2: Yes

4. Is the manuscript presented in an intelligible fashion and written in standard English?

Reviewer #1: Yes

Reviewer #2: Yes

5. Review Comments to the Author

Reviewer #1: The manuscript present interesting findings for an accurate evaluation of cotton seeds quality for improved cotton production. However, major revisions are required to make the manuscript acceptable for publication. Particularly, revisions are required with regards to the use of reference or control model for a clear comparative analysis. The following points were raised:

1- According to the authors: "The results show that the enhanced ResNet50 model can achieve an average recognition accuracy of 97.23%,approximately 5.05%, 3.16%, 2.39%,1.69%, and 1.02% better than several classical models of AlexNet, VGG16, GoogleNet, ResNet18, and ResNet50, respectively". What is the reference model or the control, and how did the it perform? The authors did not perform a clear comparative analysis among the models. How is it possible to make such conclusion?

2- Clearly state explain CBAM, LED, CDD and other abbreviations/acronyms the first time there appear in the in the manuscript.

3- Reformat tables according to PLOS ONE requirements.

Reviewer #2: The paper was insightful introducing the Impro-ResNet50 model for cotton seed identification of three types of cotton seeds: intact cotton seeds, broken cotton seeds, and cracked cotton seeds. But there were minor grammatical and typographical errors especially iin the introduction that needs attention. Also, the experimental process should be separated from the results and discussion.

6. PLOS authors have the option to publish the peer review history of their article (what does this mean?). If published, this will include your full peer review and any attached files.

Reviewer #1: **Yes: **Félicien Akohoue

Reviewer #2: No

---

## [Author Response · Author response to Decision Letter 0]

6 May 2022

Dear Editor and reviewers:

Thank you so much for your letter dated April 15. We were pleased to know that our work was rated as potentially acceptable for publication in Journal, subject to adequate revision. We acknowledge the reviewers for the time and effort that they have put into reviewing the previous version of the manuscript. Their suggestions have helped us to improve our work. Based on the instructions provided in your letter, we uploaded the file of the revised manuscript. At the same time, we have uploaded a marked-up copy of the original manuscript file, highlighting the changes made to the original version.

Appended to this letter is our point-by-point response to the comments raised by the reviewers. The comments are reproduced and our responses are given directly afterwards in a different colour (red).

We would be grateful to you for allowing us to resubmit a revised copy of the manuscript.

We hope that the revised manuscript selects for publication in the PloS ONE.

Sincerely

#Reviewer 1：

Comments to the Author

1- According to the authors: "The results show that the enhanced ResNet50 model can achieve an average recognition accuracy of 97.23%,approximately 5.05%, 3.16%, 2.39%,1.69%, and 1.02% better than several classical models of AlexNet, VGG16, GoogleNet, ResNet18, and ResNet50, respectively". What is the reference model or the control, and how did the it perform? The authors did not perform a clear comparative analysis among the models. How is it possible to make such conclusion?

Response: We have no clear comparative analysis problem for several models according to you. Regarding this point, when I re-read the manuscript, I feel that there are indeed some problems as a teacher said, the original manuscript is only expressed by a single bar graph and there are some errors in the graphs, which do not clearly represent the desired results to be shown. Therefore, I reorganized the experimental data in the revised manuscript and added the model training curve in Fig 12, the average recognition accuracy of the model in Fig 13 and the statistical values of the model training results in Table 3 for the performance comparison of several models.

Fig 12. Performance comparison between improved-ResNet50 model and different classical models.

Fig 13. Comparison of the type recognition results of several models.

Table 3. Comparison of Impro-ResNet50 performance with different classical models.

Model names Epoch Error-rate % Validation Loss Training Time /mins: secs Accuracy/%

AlexNet 300 8.621 0.3727 56.60 92.189

VGG16 300 5.9280 0.1818 357.40 94.071

GoogLeNet 300 5.1580 0.1484 158.80 94.842

ResNet18 300 4.4534 0.1402 196.40 95.546

ResNet50 300 3.7853 0.1014 208.40 96.214

Impro-ResNet50 300 2.7687 0.0827 228.40 97.231

2- Clearly state explain CBAM, LED, CDD and other abbreviations/acronyms the first time there appear in the in the manuscript.

Response: Due to personal negligence in writing the paper, the full names of the acronyms CBAM, LED, CDD, etc., were not clearly explained. CBAM: Convolutional block attention module, LED: Light-emitting diode, CCD: Charge-coupled device.

3- Reformat tables according to PLOS ONE requirements.

Response: By carefully reading the table arrangement requirements of ' PloS ONE ' and consulting recent periodical publications, all tables' titles and text format errors are corrected.

Thank you for taking the valuable time to review my article. Following teacher instructions, the article structure is more scientific and rigorous.

#Reviewer 2：

Comments to the Author

1-The paper was insightful introducing the Impro-ResNet50 model for cotton seed identification of three types of cotton seeds: intact cotton seeds, broken cotton seeds, and cracked cotton seeds. But there were minor grammatical and typographical errors especially iin the introduction that needs attention. Also, the experimental process should be separated from the results and discussion.

Response: According to your review comments, I corrected the grammatical errors and typographical errors in the introduction part and the reference citation marking problems, which I had not found carelessly. I thank you for reviewing the article carefully and pointing out the problems one by one. In addition, regarding the teacher's suggestion to separate the experimental procedure from the results and discussion, my previous paper structure was indeed not rigorous enough, and I redesigned the paper structure in the rewrite. The main structure of the thesis is shown in the figure below.

Fig.14 Main flow chart of the article.

Thank you very much for your guidance, the structure of the paper is more scientific and compact by adjusting the form of the article.

Thank you so much to the editors and reviewers for your comments on my paper, which made me benefit a lot. I am now re submitting the revised draft, and hope that the revised article will meet the requirements of the teachers, of course, if there are still problems in the paper, I hope you can be kind enough to point them out, and I will continue to improve until I meet your requirements. I hope the journal will accept my article and I look forward to listen to you.

Finally, I would like to thank the editors and reviewers again for their help.

---

## [Decision Letter · Decision Letter 1]

9 Jun 2022

PONE-D-22-02418R1Cotton seed quality identification method based on improved ResNet50 modelPLOS ONE

Dear Dr. Du,

Thank you for submitting your manuscript to PLOS ONE. After careful consideration, we feel that it has merit but does not fully meet PLOS ONE’s publication criteria as it currently stands. Therefore, we invite you to submit a revised version of the manuscript that addresses the points raised during the review process.

We look forward to receiving your revised manuscript.

Kind regards,

Sathishkumar V E

Academic Editor

PLOS ONE

Reviewers' comments:

Reviewer's Responses to Questions

**Comments to the Author**

1. If the authors have adequately addressed your comments raised in a previous round of review and you feel that this manuscript is now acceptable for publication, you may indicate that here to bypass the “Comments to the Author” section, enter your conflict of interest statement in the “Confidential to Editor” section, and submit your "Accept" recommendation.

Reviewer #1: (No Response)

Reviewer #3: (No Response)

2. Is the manuscript technically sound, and do the data support the conclusions?

Reviewer #1: Yes

Reviewer #3: (No Response)

3. Has the statistical analysis been performed appropriately and rigorously? 

Reviewer #1: Yes

Reviewer #3: (No Response)

4. Have the authors made all data underlying the findings in their manuscript fully available?

Reviewer #1: Yes

Reviewer #3: (No Response)

5. Is the manuscript presented in an intelligible fashion and written in standard English?

Reviewer #1: No

Reviewer #3: (No Response)

6. Review Comments to the Author

Reviewer #1: The authors made an effort to improve the statistical analyses and quality of the Figures and Tables. However significant improvement are still needed regarding the writing style, typography of the paper and discussion of key findings.

Abstract

“This paper proposes … convolutional block attention module(CBAM)”. Please put a space between module and CBAM. Please correct this type of error throughout the manuscript.

Introduction

Significant grammatical improvement is needed. For instance:

1. The authors said: “In the production of cotton seeds, by introducing the principles of air screen, magnetic screen, socket screen, gravity screen, dielectric screen, and other tenets of mechanized sorting equipment to replace the manual screening, significantly improving the screening efficiency of cotton seeds in actual production, this method can screen grain whole cotton seeds in large quantities.”

Which method are the authors referring to? Please reformulate the whole sentence to make it clearer.

2. Reformulate the following sentence: “Machine vision technology for the non-destructive testing application of research a lot, broad prospects, traditional machine vision technology through the linear discriminant model, artificial neural network, support vector machine and other algorithms can remove the immature cotton seed, broken cotton seed”.

3. The sentence: “In recent years, the deep learning technology represented by convolutional neural networks has been widely used in the field of image recognition; compared with the traditional image detection technology, convolutional neural networks have increased the deep network structure with feature learning, which can automatically extract the most representative image features” should be splitted into two sentences for more clarification. "Compared with ... image features." might become a separate sentence.

4. In the sentence “Researchers at home and abroad apply convolutional neural networks to seed quality detection of corn, wheat grains, peanuts and other crops”, what do the authors mean by researchers at home and abroad? Reformulate it in a scientifically sound way.

5. The authors systematically add a point (.) after most citations regardless of their position in the sentences. Some examples are “Przybyło et al [19].”, “Altuntaş et al [20].” and “Ni et al [21].” to list the least. This makes it difficult for a reader to understand the sentences.

Methodology

1. Please clearly organise this section in an appropriate manner to avoid confusion.

2. From the subsection “Model training and parameter tuning” to “Comparison of different neural network models” there was a huge confusion between methods and results. Everything relating to a result should be moved to the results section.

Results

1. “Results and Discussion”. Please delete discussion from the section title

2. Please integrate comments provided in the methodology and reorganize the results section accordingly.

Discussion

This section needs significant improvement. The authors should interpret their key results and relate each of them to previous investigations/studies in the field.

Reviewer #3: Add Literature review or related works section

Numberings for the equations are wrong

Whether hyperparameter tuning is done?

Add future works at the end of conclusion section

The contribution of the works should be added as points in the introduction section.

English language check needed

7. PLOS authors have the option to publish the peer review history of their article (what does this mean?). If published, this will include your full peer review and any attached files.

Reviewer #1: No

Reviewer #3: **Yes: **Usha Moorthy

---

## [Author Response · Author response to Decision Letter 1]

23 Jul 2022

Dear Editor and reviewers:

 Thank you so much for your letter of June 10. We are pleased that our paper "A Method for Detecting the Quality of Cotton Seeds Based on an Improved ResNet50 Model (ID: PONE-D-22-02418R1)" may be published in your journal. However, sufficient revisions are needed. We are very happy to have this opportunity. We acknowledge the reviewers for the time and effort they put into reviewing the previous version of the manuscript. With your valuable comments, I can improve my paper, which will also help me in my future work. Once again, my heartfelt thanks.

 We hope that the revised manuscript will be selected for publication in PloS ONE. I am looking forward to hearing from you.

Sincerely

#Reviewer 1：

Comments to the Author

1- Abstract

“This paper proposes … convolutional block attention module(CBAM)”. Please put a space between the module and CBAM. Please correct this type of error throughout the manuscript.

Answer: The similar error in the paper has been corrected. Thank you for your precious comments.

2-Introduction

(1) The authors said: “In the production of cotton seeds, by introducing the principles of air screen, magnetic screen, socket screen, gravity screen, dielectric screen, and other tenets of mechanized sorting equipment to replace the manual screening, significantly improving the screening efficiency of cotton seeds in actual production, this method can screen grain whole cotton seeds in large quantities.” Which method are the authors referring to? Please reformulate the whole sentence to make it clearer.

Answer: Thank you for your valuable comments. This revision has modified the introduction section significantly. Considering the logic of the context, this statement has been deleted in this revision.

(2) Reformulate the following sentence: “Machine vision technology for the non-destructive testing application of research a lot, broad prospects, traditional machine vision technology through the linear discriminant model, artificial neural network, support vector machine and other algorithms can remove the immature cotton seed, broken cotton seed.”

Answer: As you commented, this expression has a big problem. Therefore, a great deal of adjustments has been made in making changes. The partial modifications are as follows. 

 The revised sentence is “Machine learning-based image processing techniques have been successfully applied to detect seed quality with the advancement of computer vision technology [6-8]. The researchers conduct seed quality assessment by extracting features such as texture, color and shape of the seed images. This method is more advanced and effective in detecting seed quality than the manual method. However, the method is relatively dependent on manual feature extraction, and different features require different extraction methods. In addition, manual feature extraction is usually inadequate. Thus, it leads to the detection accuracy of the method is not high.”

 Thank you for your valuable comments, and I hope the revisions will meet with your approval.

(3) The sentence: “In recent years, the deep learning technology represented by convolutional neural networks has been widely used in the field of image recognition; compared with the traditional image detection technology, convolutional neural networks have increased the deep network structure with feature learning, which can automatically extract the most representative image features” should be splitted into two sentences for more clarification. "Compared with ... image features." might become a separate sentence.

Answer: As in question (2), the sentence has been changed considerably. The partial modifications are as follows. 

 The revised sentence is, “There has been an increase in convolutional neural networks used for image recognition [9-11]. In addition to simulating the human brain's mechanism for extracting features in layers, the technique can extract features automatically from simple to complex, from bottom to top, and from concrete to abstract. Several researchers have successfully applied CNN to the detection of seed quality [12-15]. But, a disadvantage of CNN detection is that it requires a large amount of training data, is time-consuming, and computationally resource-intensive.”

 Thank you again for your valuable comments.

(4) In the sentence “Researchers at home and abroad apply convolutional neural networks to seed quality detection of corn, wheat grains, peanuts and other crops,” what do the authors mean by researchers at home and abroad? Reformulate it in a scientifically sound way.

Answer: As you commented, the phrase is incorrectly phrased. I have made the following corrections. 

 The revised sentence is, “There has been an increase in convolutional neural networks used for image recognition [9-11]….. Several researchers have successfully applied CNN to the detection of seed quality [12-15]. But, a disadvantage of CNN detection is that it requires a large amount of training data, is time-consuming, and is computationally resource-intensive.”

(5) The authors systematically add a point (.) after most citations regardless of their position in the sentences. Some examples are “Przybyło et al [19].”, “Altuntaş et al [20].” and “Ni et al [21].” to list the least. This makes it difficult for a reader to understand the sentences.

Answer: According to your valuable comments, we have carefully revised these sentences in the paper. 

 The revised sentence is, “ For instance, the authors of [24] demonstrated a CNN-based transfer learning method for detecting haploid and diploid maize seeds. The model achieved optimal detection accuracy of 94.22%, providing technical support for the non-destructive, rapid, and inexpensive detection of high-quality seeds. In [25], the authors developed a peanut seed quality detection method based on machine vision and an adaptive CNN. The process achieved an average detection accuracy of 99.70% for common peanut seeds, such as mouldy, broken, or shrivelled….”

 Due to a large number of changes, please read the paper for details. Thank you again for your careful instruction.

3-Methodology.

(1) Please clearly organise this section in an appropriate manner to avoid confusion.

Answer: According to your valuable comments, we have reorganized that part of the paper. Due to a large number of changes, please read the paper for details. Thank you again for your careful instruction.

(2) From the subsection “Model training and parameter tuning” to “Comparison of different neural network models,” there was a huge confusion between methods and results. Everything relating to a result should be moved to the results section.

Answer: As you mentioned, the section is somewhat confusing. Therefore, we have made a more significant change in the paper. Due to a large number of changes, please read the paper for details. Thank you again for your careful instruction.

4-Results.

(1). “Results and Discussion.” Please delete discussion from the section title.

Answer: Thank you for your valuable comments. The section has been carefully revised. Due to a large number of changes, please read the paper for details. Thank you again for your careful instruction. 

(2). Please integrate comments provided in the methodology and reorganize the results section accordingly.

Answer: I have substantially revised that section to meet your expectations. This revision made me understand the research better. In addition, it has a vital role in improving the quality of the paper. Due to a large number of changes, please read the paper for details. Thank you again for your careful instruction. 

5-Discussion.

This section needs significant improvement. The authors should interpret their key results and relate them to previous investigations/studies in the field.

Answer: Thanks to your valuable comments, significant revisions have been made to the experimental analysis, discussion of results, etc., of the paper. This revision has helped and will continue to influence my future writing. Once again, I express my sincere gratitude. Due to a large number of changes, please read the paper for details. Thank you again for your careful instruction.

#Reviewer 3：

Comments to the Author

1- Add Literature review or related works section.

Answer: As you suggested, add a literature review or a section on related works to the paper. In addition, this revision removes older literature and adds more recent literature to increase persuasiveness.

2- Numberings for the equations are wrong.

Answer: Thank you for your careful review, and the numbering of the equations in the paper has been corrected.

3-Whether hyperparameter tuning is done?

Answer: Detailed parameter settings are performed in section 5.1 of the paper. Comparison tests with different learning rates, activation functions and fully connected layers are performed to correct the parameters in Section 6.1. Due to a large number of changes, please read the paper for details. Thank you again for your careful instruction.

4-Add future works at the end of conclusion section.

Answer: As you suggested, future work has been added at the end. The main points are the following.

 The Impro-ResNet50 cotton seed quality detection model, based on the attention mechanism, was trained on a large amount of data while maintaining high accuracy and requiring only a short amount of time to run. In the future, we will supplement the data with cotton seeds of different qualities and backgrounds so that the model has a wider range of applications. At the same time, the model is simplified so that it can be deployed on mobile and easily used by farmers.

5-The contribution of the works should be added as points in the introduction section.

Answer: As you suggested, the main contributions and innovations of the paper have been added in the introduction section as relevant. The main points are the following.

(1). Based on the appearance of defects in cotton seed, a new cotton seed dataset is created to support the development of subsequent detection algorithms.

(2). The Impro-ResNet50 model is proposed as a new method for detecting cotton seed quality based on an attention mechanism. The CBAM attention block is embedded in ResNet50 to integrate feature channel and spatial information attention and enhance the model's capacity to learn essential information about cotton seed regions.

(3). The model's application serves as a reference for developing new models, demonstrating the interoperability of deep learning models and attention mechanisms.

(4). On the basis of the cotton seed quality identification dataset, Impro-ResNet50 is subjected to extensive comparative experiments. Impro-ResNet50 is highly accurate and robust in cotton seed detection tasks, demonstrating the efficacy of the CBAM module. Provide technical support for developing cotton seed quality testing equipment in the future.

6-English language check needed

Answer: I appreciate your insightful feedback. According to your suggestion, I concentrate on revising long sentences, tenses, and repetitive descriptions in my paper. Subsequently, I have invited some colleagues proficient in English writing to review it. I sincerely hope that the revised paper will satisfy your needs.

 Thank you so much to the editors and reviewers for your comments on my paper, which benefited me greatly. I am now resubmitting the revised draft and hope that the revised article will meet the requirements of the teachers. Of course, if there are still problems in the paper, I hope you can be kind enough to point them out, and I will continue to improve until I meet your requirements. I hope the journal will accept my article, and I look forward to listening to you.

 Finally, I would like to thank the editors and reviewers again for their help.

---

## [Decision Letter · Decision Letter 2]

2 Aug 2022

A Method for Detecting the Quality of Cotton Seeds Based on an Improved ResNet50 Model

PONE-D-22-02418R2

Dear Dr. Du,

We’re pleased to inform you that your manuscript has been judged scientifically suitable for publication and will be formally accepted for publication once it meets all outstanding technical requirements.

Kind regards,

Sathishkumar V E

Academic Editor

PLOS ONE

Additional Editor Comments (optional):

Reviewers' comments:

Reviewer's Responses to Questions

**Comments to the Author**

1. If the authors have adequately addressed your comments raised in a previous round of review and you feel that this manuscript is now acceptable for publication, you may indicate that here to bypass the “Comments to the Author” section, enter your conflict of interest statement in the “Confidential to Editor” section, and submit your "Accept" recommendation.

Reviewer #3: (No Response)

2. Is the manuscript technically sound, and do the data support the conclusions?

Reviewer #3: (No Response)

3. Has the statistical analysis been performed appropriately and rigorously? 

Reviewer #3: (No Response)

4. Have the authors made all data underlying the findings in their manuscript fully available?

Reviewer #3: (No Response)

5. Is the manuscript presented in an intelligible fashion and written in standard English?

Reviewer #3: (No Response)

6. Review Comments to the Author

Reviewer #3: (No Response)

7. PLOS authors have the option to publish the peer review history of their article (what does this mean?). If published, this will include your full peer review and any attached files.

Reviewer #3: **Yes: **Usha Moorthy

---

## [Editor Report · Acceptance letter]

4 Aug 2022

PONE-D-22-02418R2 

A Method for Detecting the Quality of Cotton Seeds Based on an Improved ResNet50 Model 

Dear Dr. Du:

I'm pleased to inform you that your manuscript has been deemed suitable for publication in PLOS ONE. Congratulations! Your manuscript is now with our production department. 

Kind regards, 

on behalf of

Dr. Sathishkumar V E 

Academic Editor

PLOS ONE